# The Spatial Distribution Patterns, Physicochemical Properties, and Structural Characterization of Proteins in Oysters (*Crassostrea hongkongensis*)

**DOI:** 10.3390/foods11182820

**Published:** 2022-09-13

**Authors:** Wan Li, Ran Du, Julieth Joram Majura, Zhongqin Chen, Wenhong Cao, Chaohua Zhang, Huina Zheng, Jialong Gao, Haisheng Lin, Xiaoming Qin

**Affiliations:** 1College of Food Science and Technology, Guangdong Ocean University, Zhanjiang 524088, China; 2Guangdong Provincial Key Laboratory of Aquatic Products Processing and Safety, Guangdong Provincial Engineering Technology Research Center of Seafood, Zhanjiang 524088, China; 3National Research and Development Branch Center for Shellfish Processing (Zhanjiang), Zhanjiang 524088, China; 4Guangdong Province Engineering Laboratory for Marine Biological Products, Key Laboratory of Advanced Processing of Aquatic Product of Guangdong Higher Education Institution, Zhanjiang 524088, China; 5Collaborative Innovation Center of Seafood Deep Processing, Dalian Polytechnic University, Dalian 116034, China

**Keywords:** *Crassostrea hongkongensis*, protein, spatial distribution, physicochemical properties, structural characteristics

## Abstract

Protein content, a vital component determining the nutritional quality of oysters, is unevenly distributed in different parts of oyster. In this study, the spatial distribution (visceral mass, mantle, gill, and adductor) patterns and structural characteristics of proteins, including water–soluble proteins (WSP), salt–soluble proteins (SSP), acid–soluble proteins (ASP) and alkali–soluble proteins (ALSP) of oysters (*Crassostrea hongkongensis*) were investigated with the amino acid analyzer, circular dichroism spectroscopy (CD), fourier transform infrared spectroscopy (FTIR), and fluorescence spectroscopy. The results showed that oyster proteins were mainly distributed in the visceral mass and mantle. The protein composition was WSP, SSP, ALSP, and ASP in descending order, which conformed to the ideal amino acid pattern. Variations in secondary structure, molecular weight distribution, and thermal denaturation temperatures of the oyster proteins were observed. SSP had wider bands (16–270 kDa) than those of ASP (30–37 kDa) and ALSP (66–270 kDa). Among the four proteins, the SSP of the mantle showed the highest thermal stability (87.4 °C), while ALSP of the adductor muscle had the lowest the lowest the peak denaturation temperature (Tm) (53.8 °C). The proportions of secondary structures in oyster proteins were different, with a higher proportion of solid protein β–folds, and the exposure of aromatic amino acid residues and disulfide bonds and the microenvironment in which they were located were also different.

## 1. Introduction

Oyster (also known as sea oyster), a vital bivalve shellfish in the family Oysteridae of the phylum Mollusca, class Cladocera, order Isopoda, is the world’s leading cultured shellfish and also of great economic importance in China. According to the China Fisheries Yearbook [1], oyster mariculture production reached 5.82 million tons in 2021. *Crassostrea hongkongensis* is mainly distributed in China’s southern coastal regions of Guangxi, Guangdong, and Fujian. The edible parts of oysters contain a protein content of more than 52% (dry basis) and a balanced amino acid composition with a spectrum of essential amino acids, making them a good source of protein and known as “marine milk” [2]. Protein content plays a vital role in determining the nutritional value of food and has a significant impact on market value [2,3]. Additionally, the essential amino acid composition determines the quality of the protein [4,5]. Therefore, the nutritional value of oysters depends mainly on their proteins. 

In oysters, proteins are mainly distributed in the soft body (including visceral mass, mantle, gill, and adductor muscle). On one hand, the nutritional quality of oysters is related to the protein content of different parts of oysters. Diploidand and triploid *Crassostrea hongkongensis* are high–protein foods that maintain relatively stable levels and a relatively well–balanced composition of essential amino acids between the reproductive and non–reproductive phases. This suggests that diploid and triploid *Crassostrea hongkongensis* are high–quality sources of protein [3]. Qin et al. found significant changes in protein levels in *Crassostrea hongkongensis* between April and June, which was associated with oysters’ transition from proliferation to maturation. The oysters in the maturation phases had the highest protein content and the best quality. Therefore, the optimal harvest time of *Crassostrea hongkongensis* was during the maturation phases, roughly from August to February [2]. On the other hand, the physicochemical properties of oysters are related to the types of proteins (e.g., water–soluble proteins (WSP), salt–soluble proteins (SSP), acid–soluble proteins (ASP) and alkali–soluble proteins (ALSP) in different parts of the oyster. Jiang et al. showed that WSP in different parts of the oyster simulated gastrointestinal digestion, and the ease of digestion in descending order was visceral mass, gill, mantle, and adductor [6]. Heating resulted in significantly increased surface hydrophobicity of WSP and SSP, and decreased protein solubility. The digestion of WSP increased with the denaturation of proteins at lower heating temperatures, while protein aggregation and the formation of a more compact structure inhibited the digestion of WSP at higher heating temperatures. The digestion of SSP was inhibited by protein oxidation and aggregation [7,8]. The proteins from different parts of the oyster (visceral mass, mantle, gill, and adductor) had different abilities to bind metals such as Cd, Zn, and Cu, indicating that metal-binding ability correlates with the protein type [9,10,11,12]. Oyster proteins (e.g., WSP, SSP, ASP and ALSP) have different physicochemical, functional, and structural properties, and are suitable for different processing methods [7,8,13,14,15,16,17]. 

Muscle proteins of oysters can be divided into WSP (myoplasmic proteins), SSP (myofibrillar proteins), ASP, and ALSP based on differences in solubility. Unlike most shellfish whose muscle protein composition were dominated by SSP, oysters are dominated by WSP followed by SSP [7,18,19]. WSP have a complex variety of composition, a small molecular weight (about 1.0 × 10^4^–3.0 × 10^4^), are all close to spherical in shape, and contain many enzymes related to metabolism. SSP are structural proteins that support muscle movement and are composed mainly of about 50% myosin, 20% actin, 5% promyosin, and troponin. The relative molecular masses of myosin and actin are about 5.0 × 10^5^ and 4.5 × 10^4^, respectively. ASP and ALSP are parts of the muscle matrix proteins, including collagen and elastin, which are the main components of connective tissue. Elastin constitutes a smaller proportion of the connective tissue of skeletal muscle (only 0.5%) and has less impact on processing and utilization [7,20,21].

This study aimed to investigate the nutritional value and structural properties of the proteins isolated from oyster *C. hongkongensis*. The proteins of interest included WSP, SSP, ASP, and ALSP, which were obtained from several parts of the soft oyster body (visceral mass, mantle, gill, and adductor) and separated based on their solubility differences. The structural characteristics of WSP, SSP, ASP, and ALSP were examined by CD, FTIR, and fluorescence spectroscopy. Overall, the findings will provide an essential reference for the processing and preservation of oysters and explore new ways to develop and apply high–value–added oysters. 

## 2. Materials and Methods

### 2.1. Materials 

Oysters (length (12.8 ± 0.6) cm, width (6.0 ± 0.4) cm) were bought from a local aquatic market in Zhanjiang (Guangdong province, China) and immediately transported to the laboratory. SDS–PAGE gels kit, SDS–PAGE electrophoresis solution (Tris–Gly, 10×), SDS–PAGE protein loading buffer, DS–PAGE premixed protein marker, BeyoBlue Coomassie brilliant blue ultrafast staining solution were purchased from Beyotime Biotechnology Co., Ltd. (Shanghai, China). All of the reagents used in the study were of analytical grade. 

### 2.2. Extraction of Proteins 

The oysters were dissected into four parts (viscera mass, mantle, gill, and adductor) using tweezers, scissors, and a scalpel (Appendix A). Each tissue was frozen in liquid nitrogen, weighed, and stored at −80 °C. WSP, SSP, ASP, and ALSP were prepared based on their solubility differences. WSP was further prepared according to Zhang et al. with slight modifications [7]. Briefly, 25 g of material was suspended in a 200 mL buffer solution (50 mM PBS pH 7.2) and stirred for 90 min at 12 °C. The obtained mixture was then centrifuged at 12,000× *g* rpm, 4 °C for 20 min (TDL–5–A, Shanghai, China), and the precipitate was repeatedly extracted three times, the supernatant was combined, and the supernatant was the WSP. A total of 200 mL of supernatant was added to 200 mL of 10% trichloroacetic acid and left for 30 min. The mixture was centrifuged at 12,000× *g* rpm for 20 min at 4 °C to take the precipitate for WSP content determination. SSP was prepared according to Zhang et al. with slight modifications [8]. Four times the volume of 0.1 M PBS (pH 7.2, 0.5 M NaCl) was added to the remaining precipitate after extraction of the water-soluble proteins, and then stirred at 4 °C for 18 h. The mixture was centrifuged at 12,000× *g* rpm for 20 min (4 °C) and the supernatant was collected. The extraction steps were repeated three times. The three supernatants were combined and dialyzed with the 3.5 kDa cut-off dialysis tubing. After that, all of the supernatants were lyophilized to obtain the protein powder. The extraction process was conducted three times. The precipitation was rinsed with water thrice and then soaked in 0.1 M HCl or NaOH to extract ASP and ALSP. The protein components were desalted and freeze-dried for later use [22]. 

### 2.3. Amino Acid Analysis

The amino acid compositions of WSP, SSP, ASP, and ALSP from the parts mentioned earlier were analyzed according to Jiang et al. [6]. Briefly, 10 mg of protein was placed in a 10 mL ampoule and mixed with 5 mL of 6 M HCl before being sealed, followed by hydrolysis at 110 °C for 24 h in an air oven. Next, 5 mL of the solution was evaporated under a stream of nitrogen. The dried samples were dissolved in a 5 mL sodium citrate buffer (pH 2.2) and loaded on an automatic amino acid analyzer. The tryptophan contents of the protein were determined by HPLC (Agilent 1200, Shimadzu Ltd., Tokyo, Japan) after alkaline hydrolysis. Finally, the amino acid composition was expressed as mg/100 g protein. The reference pattern of amino acids was taken from FAO/WHO (1973) [23]. The EAA score (EAAS) was calculated as follows:$$\mathrm{EAAS}=\frac{\mathrm{mg}\;\mathrm{of}\;\mathrm{EAA}\;\mathrm{in}\;g\;\mathrm{protein}\;\mathrm{of}\;\mathrm{test}\;\mathrm{samples}}{\mathrm{mg}\;\mathrm{of}\;\mathrm{EAA}\;\mathrm{in}\;g\;\mathrm{protein}\;\mathrm{of}\;\mathrm{FAO}/\mathrm{WHO}\;\mathrm{referencepattern}}\times 100$$

### 2.4. Sodium Dodecyl Sulfate-Polyacrylamide gel Electrophoresis (SDS-PAGE) 

The SDS–PAGE analyses of proteins were performed with 12% separating gel and 5% stacking gel [24]. A total of 10 μg of protein was loaded into each well. A standard protein marker with molecular sizes ranging from 16 to 270 kDa was used for molecular mass confirmation. Electrophoresis was carried out at a constant voltage of 120 V for about 2 h.

### 2.5. Differential Scanning Calorimetry (DSC) Analysis

The thermal properties of the proteins were investigated using a previously described method [7]. The experiment was conducted using differential scanning calorimetry–DSC (204F1 Phoenix, Schneider, Germany) at a scanning rate of 10 °C/min in the temperature range of 30–120 °C. 

### 2.6. UV Spectroscopy

The UV–vis spectroscopy (Cary 60 UV–Vis, Agilent, USA) measurements of protein from various parts of the oyster (0.15 mg/mL) were done using the spectrophotometer with a 1 cm cuvette at room temperature. The scan range was 190–400 nm, and the scan speed was 5 nm/s [7].

### 2.7. Fourier Transform–Infrared Spectroscopy (FTIR) Analysis

Fourier transform infrared (FTIR) spectra (BRUKER TENSOR 27, Bruker, Germany) of the oyster proteins were obtained from KBr discs. Each disc contained an approximately 1.0 mg sample and 100 mg KBr. The spectra were recorded in the 400–4000 cm^−1^ range on a Spectrum One FTIR Spectrophotometer [25]. Fourier self–deconvolution was conducted to analyze protein secondary structure by employing the amide I region (1700–1600 cm^−1^). 

### 2.8. Circular Dichroism (CD) Analysis

To investigate the secondary structure of proteins (α–helix, β–fold, β–turn, and random coil) based on the previous report with slight modifications [26], CD measurements of the samples were conducted to investigate the secondary structure of proteins (α–helix, β–fold, β–turn, and random coil) with a Chirascan CD spectrophotometer (Chirascan™, Applied Photophysics Ltd., Leatherhead, England) between 190 and 280 nm with a bandwidth of 1 nm, a scanning rate of 100 nm/min, and a 1 nm data pitch. Each spectrum was measured against an appropriate reference with eight scans accumulation for each sample. The protein concentration used was 0.2 μg/mL. The distilled water background was subtracted. Protein secondary structural analysis of the CD spectra (190–260 nm) was conducted using CDNN 2.1.0. 

### 2.9. Fluorescence Spectroscopy Analysis

The fluorescence was determined using the method described by Wang et al. with slight modifications [27]. Fluorescence measurements were obtained with an RF–5301PC spectrofluorimeter (Shimadzu, Japan). The protein concentration was 0.15 mg/mL. Excitation–emission spectra were measured at 25 °C and processed to obtain the emission spectra (λ_em_: 300–450 nm) at the maximum excitation wavelength (292 nm). The excitation and emission slit widths were 5.0 nm. All measurements were performed in triplicate. Synchronous fluorescence was determined using the method of Gao et al. with slight modifications [28]. Synchronous fluorescence spectra of proteins from different parts of oysters were recorded at λ_em_ = 230–450 nm with Δλ of 15 and 60 nm. The excitation and emission bandwidths were 5.0 nm. 

### 2.10. Statistical Analysis 

The results were expressed as mean values ± standard deviations (SD) and analyzed using one–way analysis of variance (ANOVA) to estimate the significance of differences among mean values. A probability value of *p* < 0.05 was taken to indicate statistical significance. 

## 3. Results and Discussion

### 3.1. Protein Composition Analysis 

Protein content determines the nutritional value of foods and substantially impacts the food’s market value [2,3]. Oysters are a good protein source, with abundant amounts in the visceral mass, mantle, gill, and adductor muscle (Appendix A). Figure 1 depicts the protein composition derived from these parts. Among the selected regions, the highest protein content was observed in the adductor muscle (*p* < 0.05) (Figure 1A). The proportion and protein content trends from several parts were significantly different (Figure 1A,B). The percentage of protein in visceral mass was the highest (*p* < 0.05) at 38.23%, followed by mantle, adductor, and gill (the lowest). The protein extraction rates were, in descending order, 85.19%, 80.74%, 75.47%, and 66.48% for the adductor, mantle, gill, and visceral mass, respectively (Figure 1C).

WSP, SSP, ASP, and ALSP were the most common proteins. The contents of WSP, SSP, ASP and ALSP varied widely in different parts of oyster, as well as between different proteins in the same part. The contents of the different proteins in the visceral mass and the mantle were WSP, SSP, ALSP and ASP in descending order. The content of SSP was higher than that of WSP in the adductor, while the contents of ASP and ALSP were lower. The ALSP of the gill was 30.93%, which was significantly higher than the other parts. For the gill, the major fraction was ALSP, followed by WSP, SSP, and ASP, respectively (Figure 1C). Karnjanapratum et al. showed that content of ALSP in *Meretrix lusoria* was the highest, followed by WSP, SSP, and the lowest was stroma protein [21]. WSP, also known as sarcoplasmic protein, has good biological activity due to its small relative molecular weight and good solubility. At the same time, WSP contain a lot of cathepsins, which leads to their easy degradation. WSP also have a great influence on the texture and color of meat [29]. The contents of WSP in the different parts of oyster from high to low was the mantle, gill, visceral mass, and adductor. Jiang et al. showed that WSP in different parts of the Pacific oyster (visceral mass, mantle, gill, and adductor) varied widely. In simulated gastrointestinal digestion, visceral mass protein had the highest digestibility, followed by gill, mantle, and adductor protein [6]. SSP, also known as myogenic fibrillar protein, is abundant in muscle and is closely associated with water-holding, emulsifying and gel properties of meat [20]. SSP was the most abundant in the adductor, followed by the mantle, visceral mass and gill. ASP and ALSP are components of the muscle matrix proteins, which consist mainly of collagen and elastin and are the main components of connective tissue and are associated with the tenderness and elasticity of the meat. Since elastin is highly insoluble, ASP and ALSP are mainly collagen.

### 3.2. Amino Acid Analysis 

Protein is an essential nutrient for the human body, as it is involved in the structure and physiological functioning of the body. One of the significant aspects in determining the nutritional quality of protein is its essential amino acid composition [4,30]. As shown in Appendix A, the findings revealed that each part of the oyster protein contains 18 amino acids, including all essential amino acids (EAA) and sulfur-containing amino acids. Except for ALSP in the gill, the 15 proteins in different parts of the oyster had the highest glutamic acid content, ranging from 123–191 mg/g of protein. SSP contained more Glu than WSP, ASP, and ALSP. Studies have shown that several meat types: beef, pork, lamb, and poultry are high in Asp and Glu [31,32]. Glu has an essential function in regulating the human immune system as an excitatory neurotransmitter and a substrate for the formation of g-aminobutyric acid (GABA) in the neurological system, lymphocytes, and macrophages [33]. Asp content in SSP and ASP of adductor, visceral mass proteins, and mantle proteins was slightly lower than Glu. Asp affects cellular metabolism, gene expression, and immunity [34]. The WSP, SSP, and ASP of the gill only had lower Trp content than Glu, and the ALSP of gill had higher Trp content than Glu. Trp and Tyr create the most substantial fluorescence emission peaks in the wavelength range of 330–350 nm [24]. As a result, Trp is primarily responsible for the fluorescence emission peak of gill proteins at 330–350 nm. The primary amino acids of the proteins extracted from different parts of the oyster include Glu, Asp, Arg, Leu, and Lys, which are similar to the results of previous studies [35]. Sweet, bitter, sour, salty, and umami flavors are related to amino acids and essential components in increasing food taste [36]. The flavor amino acids include Asp, Glu, Gly, Ala, Tyr, and Phe. Oyster proteins contain up to 40%–50% of flavor amino acids, suggesting that oysters have a more pleasant flavor. The hydrophobic amino acids (HAA) are commonly seen in peptides with immunomodulatory actions [37,38]. The protein extracted from gill possesses more HAA than other proteins.

According to the FAO/WHO (1973) recommended model, the ratio of essential amino acids to total amino acids (EAA/TAA) is about 40% and the ratio of essential amino acids to non-essential amino acids (EAA/NEAA) is more than 60% for proteins of good quality [23]. Except for WSP and ASP in visceral mass, ASP in the mantle, and ALSP in the adductor, all other proteins from various parts of the oyster conformed to the ideal protein model recommended by FAO/WHO (1973). In addition, proteins of the gill were of higher quality compared to other parts, with EAA/TAAs of 45.57%, 45.72%, 41.20% and 44.98% for WSP, SSP, ASP and ALSP, respectively; and EAA/NEAAs of 83.71%, 84.21%, 70.06% and 81.77%. 

The amino acid score is an important indicator of the nutritional quality of proteins. As shown in Table 1, all the extracted proteins were rich in essential amino acids. These results align with similar results on the protein composition of several protein fractions of the Pacific oyster [6]. It is noteworthy that the content of Try in proteins of the gill WSP, and the SSP of the adductor was almost 7–13 times higher than the FAO/WHO (1973) model. The first limiting amino acid of the oyster proteins was the presence of sulfur amino acids, which is consistent with results from of the shrimp head (*Penaeus vannamei*) [30]. The EAA content of oyster proteins is similar to that of the Pacific oyster (about 392 mg/g protein) and much higher than that of shrimp head (340.29 mg/g protein) [6,30]. The amino acid composition analysis showed that oyster protein was rich in essential amino acids, flavoring amino acids, and hydrophobic amino acids and thus is a potential protein reservoir.

### 3.3. The Molecular Weight Distribution of Proteins 

SDS–PAGE characterized the molecular weight (MW) distribution of the oyster proteins, and the MW varied from 16–270 kDa (Figure 2). The WSP in different parts of the oyster had at least six protein bands, and the MW of WSP in the adductor was high and widely distributed, mainly in the range of 270, 50, 16–10 kDa. Fewer bands and lower MW were observed in proteins of the adductor, visceral mass, mantle, and gill. There was a dark and wide protein band at 37–30 kDa (Figure 2A). The MW distribution of WSP was 96, 64.5, 37–25, and 16–12 kDa, corresponding to leucine aminopeptidase (LAP), hemoglobin, cathepsin, myoglobin (Mb), and albumen, respectively (Table 2). The MW of WSP in the oyster was low, mainly distributed in the range of 63–25 kDa [7]. The MW of SSP in different parts of the oyster are shown in Figure 2B. The MW distribution of SSP was wider (16–270 kDa). The broad range can be attributed to the presence of more than one type of protein. The fraction may contain not only the large protein myosin, but also smaller ones (Figure 2B). The MW of the mantle protein was mainly distributed between 37–52 and 95–120 kDa, with few and darker protein bands. The protein bands of the visceral mass and gill were similar to the MW distribution between 16–200 kDa. The MW of the adductor was mainly distributed in 66–200kDa and 37–52kDa, and the protein bands were wide and dark in color. The bands of SSP in oyster at 200, 100, 48 and 16 kDa correspond to myosin heavy chain (MHC), paramyosin (PM), actin (A), and myosin light chain (MYL) [6,18]. The SSP of the oyster accumulated in the vicinity of 245 kDa [8]. Myosin, which accounts for about 30% of the total muscle protein and 55% of myofibrillar protein, is the most important protein for the quality of meat products [39]. A myosin molecule consists of two MHCs and four MYLs, with MW of 200 kDa and 16 kDa for MHC and MYL, respectively. The MW distribution of ASP and ALSP were shown in Figure 2C,D). Fewer bands with a pale color were observed, especially for visceral mass and gill. 

### 3.4. The Thermal Stability Analysis 

Heat treatment is the most utilized processing technology in food production and processing. The sensory, physical, and chemical qualities and the nutritional taste of aquatic products are all affected by heat treatment [40]. Excessive heating can cause a rapid loss of nutrients and water, muscle atrophy, and texture hardening, all of which can degrade the quality of aquatic products. Understanding the heat stability of proteins is therefore essential for effective protein use. The aggregation of molecules generates thermal denaturation through covalent, non–covalent, and hydrophobic interactions, and denaturation temperature is a measure index of the thermal stability of proteins. 

The DSC thermal analysis profiles of the WSP, SSP, ASP, and ALSP were shown in Figure 3. There was only one absorption peak for all the proteins in different parts of the oyster, and the protein denaturation temperature started at 35 °C, and the peak denaturation temperature (Tm) was in the range of 50–90 °C. Tm was proportional to the stability of the protein. The SSP of the mantle had the highest Tm (87.4 °C), and the ALSP of the adductor had the lowest Tm (53.8 °C). The SSP of the mantle may be more structurally stable than the other proteins, while the ALSP of the adductor may be most susceptible to denaturation. For the Tm of the SSP, the mantle was the highest, followed by the gill, viscera mass and adductor, respectively (Figure 3B). The denaturation temperature range of SSP in the oyster was 35–60 °C [8]. The Tm of WSP was 67 °C for the viscera mass, gill and adductor, and a relatively high Tm of 73.4 °C for the mantle (Figure 3A). One study showed that the Tm of WSP in the oyster was 39.3 °C [7]. The Tm of ASP in different parts of the oyster did not differ much, and the Tm of ALSP was from low to high in the adductor, mantle, visceral mass, and gill (Figure 3C,D).

Thermal denaturation of proteins is one of the most common denaturation processes. After heating, changes in the spatial structure of proteins (excluding primary structures) cause changes in their function and characteristics [8]. The fundamental explanation for the difference in protein quality and digestibility following heat treatment is the change in the protein structure [41,42,43]. Thermal denaturation has been found to alter the secondary structure of proteins significantly. The mantle is placed at the outer edge of the entire visceral mass and has a comparatively thin structure. During heat treatment, the heat transport time from the surface to the interior is quick and protein denaturation is severe. The gill has a similar structure to the mantle: both are lamellar and have a large heating area and both are positioned on the surface of the entire organ of the oyster and transmit heat quickly. The leading causes of oysters’ bad taste are both of these aspects. The hardness of the mantles and gills rises while the elasticity of the gill reduces during the heating process, resulting in a more rigid texture of the oyster. 

### 3.5. Secondary Structure of Proteins in Different Parts of the Oyster 

Protein structure is closely related to its functional properties, and protein secondary structure is the basis of the spatial conformation of protein complexes. Protein secondary structure refers to the local spatial structure of the polypeptide chain backbone, and the common secondary structures include regular structures such as α–helix, β–fold, and irregular structures such as β–turn and random coil. Hydrogen bonds mainly maintain the various secondary structures, but other forces such as van der Waals, coordination, and disulfide also contribute [44]. The oyster proteins circular dichroism and protein secondary structure content were evaluated using far–UV circular dichroism spectroscopy. The circular dichroism and secondary structure of protein solutions are indicated in Figure 4. The proteins contained α–helix, β–fold, β–turn, and random coil structures in different percentages. Except for ASP of the mantle, the same proteins of the visceral mass, mantle, and gill all have equal portions of secondary structure. For the adductor, its WSP, SSP, ASP, and ALSP had a higher percentage of α–helix and the lowest percentage of β–fold, and the percentage of secondary structures (α–helix, β–fold, β–turn, and random coil) of adductor protein differed from other proteins.

In addition to circular dichroism, infrared spectroscopy can also be used to analyze the secondary structure of proteins. These two techniques complement each other to achieve an accurate description of the secondary structure of proteins [45]. In general, the amide I bands in IR spectra are the most valuable for studying protein secondary structure [46]. The relationship between the amide I band split peak fitting, and secondary structure is as follows: β–fold for absorption peaks 1617–1623 cm^−1^ and 1691–1698 cm^−1^, β–turn for absorption peaks 1667–1685 cm^−1^, random coil for absorption peaks 1636–1643 cm^−1^, and α-helix for absorption peaks 1647–1658 cm^−1^ [47]. The content of each secondary structure of the proteins could be calculated using split–peak fitting. The infrared absorption profiles of oyster protein powder at 400–4000 cm^−1^ are seen in Figure 5A–D. It is similar to the infrared spectra of the Pacific oyster protein [6]. The absorption peaks of protein and polysaccharides vibrate in the region from 1600 to 1700 cm^−1^, and the absorption peaks are due to the presence of C=O and weak C–N stretching and N–H bending. The fitted fractional peaks of oyster proteins were widely distributed between 1600 and 1700 cm^−1^.

The α–helix, β–fold, β–turn, and random coil structures found in oyster protein are shown in Figure 5Aa–Dd. Higher proportions of β–fold of proteins were observed. Except for ASP of gill and ALSP of adductor, the proportion of α–helix, β–turn, and the random coil of ASP and ALSP from several parts were α–helix, random coil, and β–turn from high to low. The proportion of α–helix, β–turn, and the random coil of WSP and SSP were more diverse than in other proteins. In WSP, the lowest proportion of random coil structure was in the adductor, followed by visceral mass. In SSP, the visceral mass did not contain α-helix structure. 

As with the pulse proteins, the oysters’ protein powder from the selected parts has a larger β–fold, better thermal stability, and a higher denaturation temperature. The predominance of β–sheets in the secondary structure also contributes to lowering the digestibility of pulse proteins [48]. A high linear negative correlation between food digestibility and β–fold content has been suggested, and the random coil, in contrast to the β–fold, promotes protein digestion [44,49,50]. β–fold content increased, and digestibility decreased after heat treatment of oyster water–soluble proteins [29,50]. It has been found that microwave treatment of shrimp proteins with increased β–fold content resulted in decreased protein digestibility in vitro [51]. Heat and pressure-treated oyster proteins have less α–helix and β–sheet, increasing protein solubility and digestibility [49]. The content of α-helix in ball–milled oyster protein increased significantly (*p* < 0.05), which suggested that the secondary structure of oyster protein was more stable and orderly after ball-milling treatment [15]. Changes in protein structure affect its functional properties and digestibility [51,52]. 

Therefore, the percentages of secondary structures (α–helix, β–fold, β–turn, and random coil) of proteins in different parts of oyster in liquid and solid states differed significantly. The percentage of the random coil in the liquid state was higher than that in the solid state. Except for the WSP of the mantle and the WSP of the gill, the portion of β–fold was lower than that in the solid–state, which had higher digestibility and reduced thermal stability and denaturation temperature than that in the solid state. The α–helix and β–fold percentages of ASP of visceral mass and ASP of gill in the liquid state were lower than those of other proteins. The stability and functional properties of proteins are influenced by their structural properties (e.g., β–fold, random coil). As a result, this research provides valuable information for the selection of appropriate processing methods (e.g., heat treatment, enzymatic treatment, high–pressure treatment) and processing conditions (e.g., temperature, pressure, time) for various parts of the product. As filter feeders, the gills of oysters tend to accumulate bacteria and trace metal ions. Oysters can be consumed and stored by removing the gills to extend shelf life and avoid excessive metal overload. At the same time, gill metal-binding proteins, such as zinc–binding proteins, are isolated to produce valuable products to prevent capital waste and maximize its added value.

### 3.6. Conformation of Proteins in Different Parts of Oyster 

UV and fluorescence spectroscopy detect protein macromolecules and the microenvironmental polarity within them [53]. The UV absorption spectra of proteins in different parts of the oyster exhibited two characteristic protein peaks (Appendix A), The absorption peak at 200 nm reflected the side chains of amino acids (particularly Tyr, Trp, Phe, His, and Met), mainly peptide bonds (amide chromophore). The absorption peak at 280 nm reflected chromophores composed of aromatic amino acid residues (Trp, Tyr, and Phe) and disulfide bonds [49]. The maximum absorption peak of proteins in different parts of the oyster at 200 nm indicated that the peptide bonds were the major absorbing group in the ultraviolet region. At the same concentration of protein, except for the SSP of adductor and the WSP and SSP in different parts of oyster, ALSP of visceral mass had more aromatic amino acid residues and disulfide bonds exposed on the protein surface than the ASP in other parts of oyster, the ALSP of the mantle, gill, and adductor. For WSP, the mantle, gill, adductor, and visceral mass were from the highest to the lowest, and the SSP were gill, visceral mass, mantle, and adductor from the highest to the lowest. 

Fluorescence spectroscopy mainly studies the microenvironmental polarity of aromatic amino acid residues (Trp, Tyr, and Phe) [7]. The maximum emission wavelengths of Trp, Tyr, and Phe residues in protein are 348, 303, and 282 nm, respectively. In synchronous fluorescence, the fluorescence spectrum at Δλ = 15 nm reflects the spectral characteristics of protein tyrosine residues, and the spectral characteristics of Trp residues are obtained at Δλ = 60 nm. The microenvironment and the amount of aromatic amino acid residues exposed to the protein surface differed in proteins in different oyster parts (Figure 6). The amount of aromatic amino acid residues (Trp, Tyr, and Phe) exposed to the protein surface differed in proteins from selected parts of the oyster, and there were differences in the significant aromatic amino acid residues of fluorescent origin (Figure 6A–D). In synchronous fluorescence, the Tyr residues of WSP in gill were exposed the most on the protein surface, and the Trp residues of SSP in visceral mass mainly were exposed on the protein surface. The Tyr and Trp of proteins in different parts of the oyster were in different microenvironments (Figure 6A1–D2). The maximum wavelength of WSP and SSP fluorescence emission in oysters was the same, about 338 nm [8,29], which was consistent with the SSP in different parts of the oyster. 

## 4. Conclusions

The protein content was at its maximum in the visceral mass followed by the mantle, adductor, and gill. WSP, SSP, and ALSP were abundant proteins in oysters, rich in essential amino acids, flavor amino acids, and hydrophobic amino acids. The molecular weight distribution of oyster proteins was different: the molecular weight distribution of SSP was wider (16–270 kDa), the bands of ASP and ALSP were fewer (ASP mainly concentrated in 30–37 kDa, ALSP primarily dense in 66–270 kDa), Tm was between 50–90 °C, and ALSP of the adductor had the lowest Tm (53.8 °C), whereas SSP of the mantle had the highest Tm (87.4 °C). The percentage of secondary structures (α–helix, β–fold, β–turn, and random coil) of liquid–state and solid–state proteins in different parts of oyster differed significantly. The content of aromatic amino acids varies greatly, and the amount of aromatic amino acid residues exposed to the protein surface and the microenvironment in which they were located vary. This study was conducted to explore the correlation between protein characteristics and their edible qualities through the study of several different protein characteristics in different parts of the oyster to enrich the basic research data of aquatic products in China and to provide relevant theoretical support and practical basis for the deep processing of oysters in the future. 

## Figures and Tables

**Figure 1 foods-11-02820-f001:**
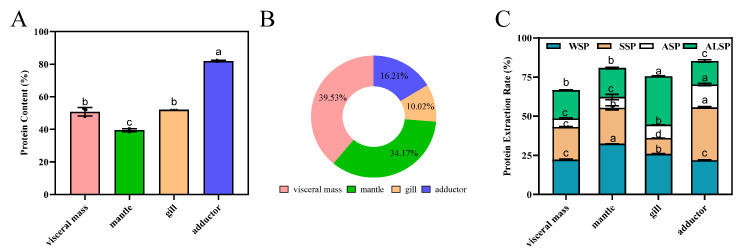
Composition of proteins from different parts of the oyster. (**A**): Protein content of visceral mass, mantle, gill, and adductor of the oyster. Different superscripts (a, b, c) indicate significant differences (*p* < 0.05). (**B**): Proportion of protein content in oyster tissues. (**C**): Extraction rate of water-soluble protein (WSP), salt-soluble protein (SSP), acid-soluble protein (ASP), and alkali-soluble protein (ALSP) in oyster tissues, protein extraction rate (%) = protein mass in the extracted sample/total protein mass of the sample × 100%. Different superscripts (a–c) in the same color indicate significant differences (*p* < 0.05).

**Figure 2 foods-11-02820-f002:**
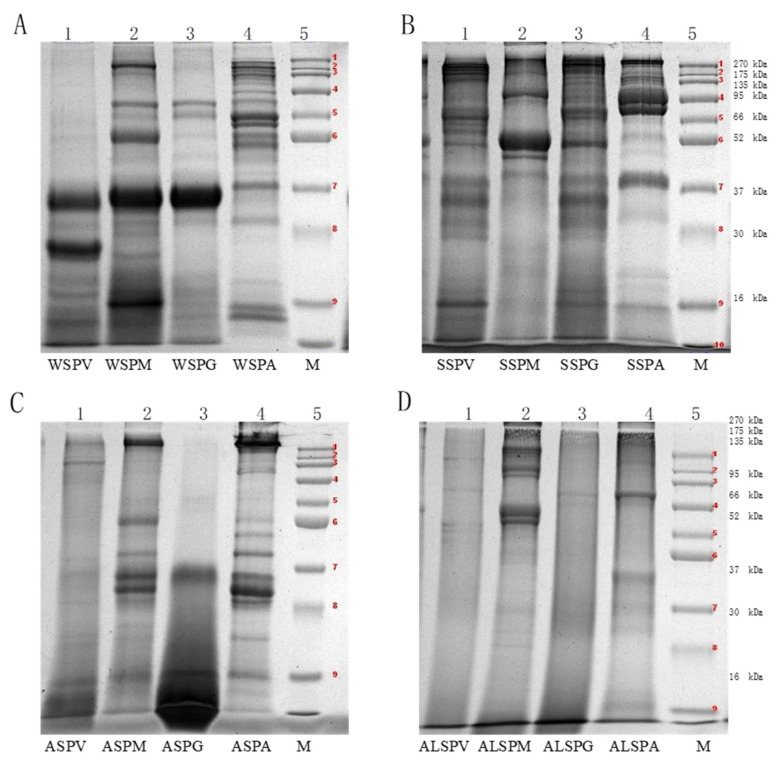
Molecular mass distribution of proteins isolated from different parts of oyster (*Crassostrea hongkongensis*). (**A**), Electrophoregrams of the WSP components in visceral mass, mantle, gill and adductor (1–WSPV, 2–WSPM, 3–WSPG, 4–WSPA, 5–protein standard). (**B**), Electrophoregrams of the SSP components in visceral mass, mantle, gill and adductor (1–SSPV, 2–SSPM, 3–SSPG, 4–SSPA, 5–protein standard). (**C**), Electrophoregrams of the ASP components in visceral mass, mantle, gill and adductor (1–ASPV, 2–ASPM, 3–ASPG, 4–ASPA, 5–protein standard). (**D**), Electrophoretogram of the ALSP components of visceral mass, mantle, gill and adductor (1–ALSPV, 2–ALSPM, 3–ALSPG, 4–ALSPA, 5–protein standard). Protein bands 1-9 of the protein marker were 270, 175, 135, 95, 66, 52, 37, 30 and 16 kDa proteins, respectively.

**Figure 3 foods-11-02820-f003:**
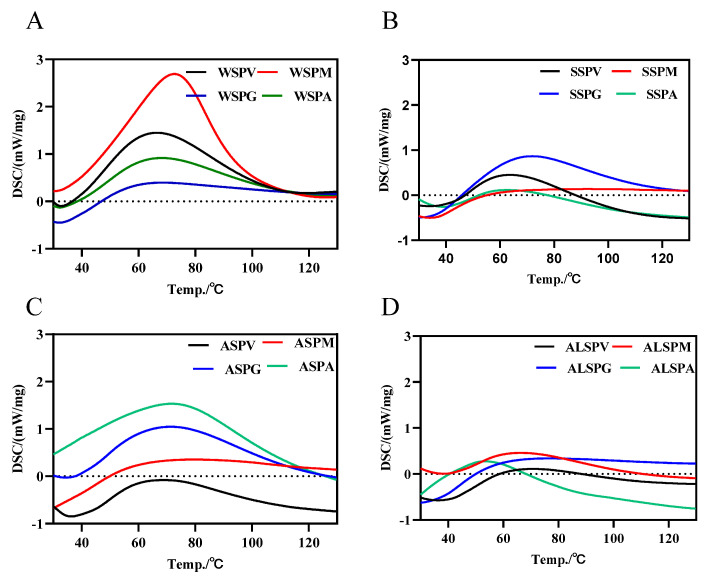
DSC profiles of protein components in different parts of the oyster. (**A**), DSC profile of WSP in different parts of oyster. (**B**), DSC profile of SSP in different parts of oyster. (**C**), DSC profile of ASP in different parts of oyster. (**D**), DSC profile of ALSP in different parts of oyster. WSPV, WSPM, WSPG, and WSPA are water–soluble proteins of the visceral mass, mantle, and adductor, respectively. SSPV, SSPM, SSPG, and SSPA are salt–soluble proteins of visceral mass, mantle, and adductor muscle, respectively. ASPV, ASPM, ASPG, and ASPA are acid–soluble proteins of the visceral mass, mantle, and adductor muscle, respectively. ALSPV, ALSPM, ALSPG, and ALSPA are alkali-soluble proteins of visceral mass, mantle, and adductor muscle, respectively.

**Figure 4 foods-11-02820-f004:**
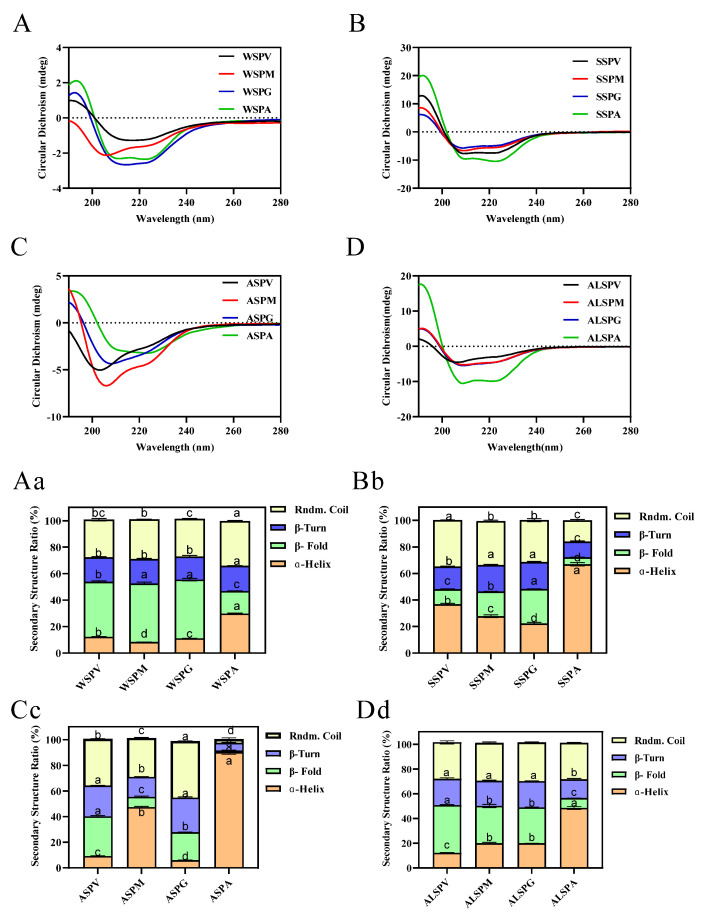
Far–UV CD spectra and secondary structure analysis of proteins in visceral mass, mantle, gill, and adductor. Figure 4 (**A**–**D**) showed the far–UV CD spectras of WSP, SSP, ASP and ALSP in different parts of oyster, respectively. Figure 4 (**Aa**–**Dd**) showed the secondary structure profiles of WSP, SSP, ASP and ALSP in different parts of oyster, respectively. Different superscripts in the same color indicate significant differences (*p* < 0.05).

**Figure 5 foods-11-02820-f005:**
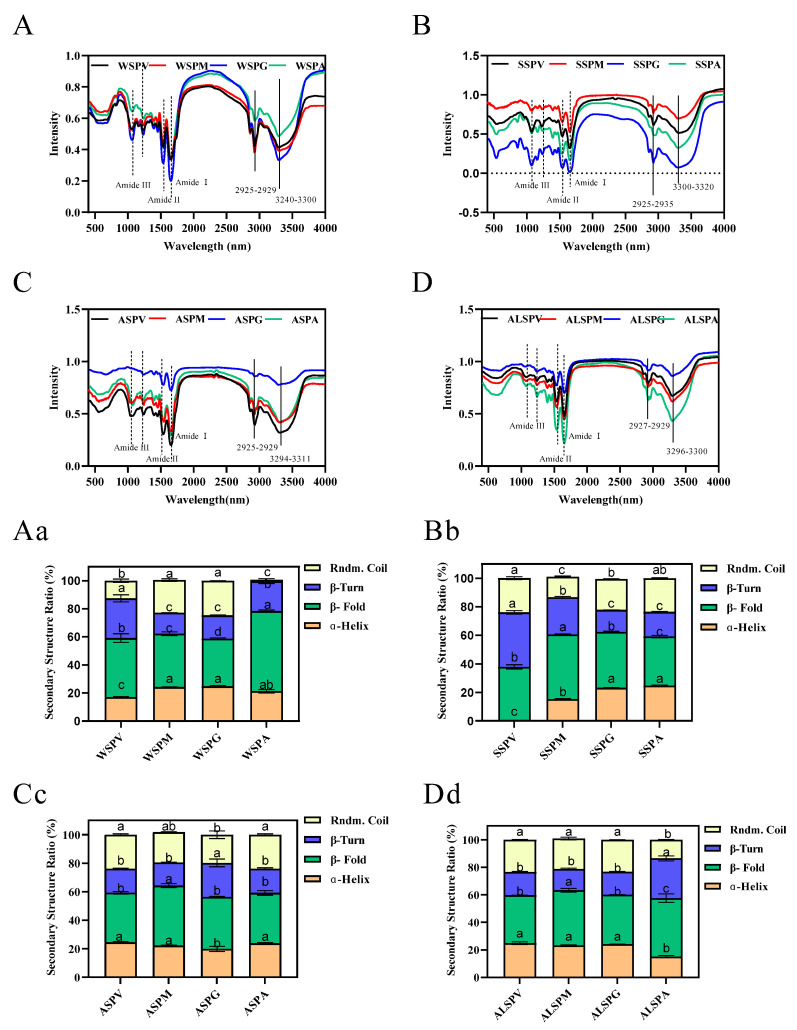
FTIR spectra and secondary structure of oyster proteins extracted from various parts. Figure 5 (**A**–**D**) showed the FTIR spectras of WSP, SSP, ASP and ALSP in different parts of oyster, respectively. Figure 5 (**Aa**–**Dd**) showed the secondary structure profiles of WSP, SSP, ASP and ALSP in different parts of oyster, respectively. Different superscripts in the same color indicate the significant differences (*p* < 0.05).

**Figure 6 foods-11-02820-f006:**
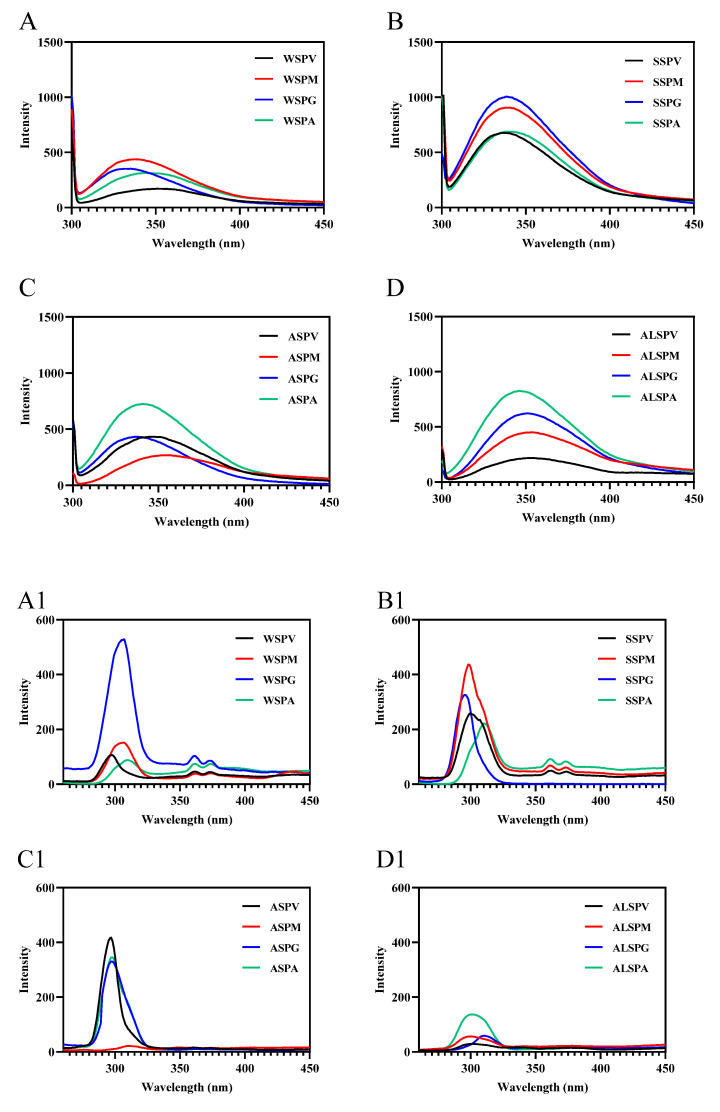
Fluorescence spectra of proteins from different parts of the oyster. (**A**–**D**), emission spectra of oyster proteins from various parts; (**A1**–**D1**), synchronous fluorescence spectra (Δλ = 15); (**A2**–**D2**), synchronous fluorescence spectra (Δλ = 60).

**Table 1 foods-11-02820-t001:** Essential amino acid score (EAAS) of proteins in different parts of oyster (*Crassostrea hongkongensis*).

Amino Acids	Reference (mg/g Protein)	WSP				SSP				ASP				ALSP			
		V	M	G	A	V	M	G	A	V	M	G	A	V	M	G	A
Val	50.00	122.44	119.73	83.20	80.70	120.05	95.73	87.54	74.33	90.99	92.51	79.17	91.29	100.78	103.78	81.86	74.94
Met+Cys	35.00	66.85	72.51	87.99	30.41	74.10	65.24	89.12	34.14	61.02	50.54	82.42	65.39	74.09	67.51	71.74	82.62
Ile	40.00	97.49	113.88	98.36	105.24	114.17	124.78	102.72	100.32	114.37	108.75	88.66	133.71	127.08	126.64	94.16	96.00
Leu	70.00	83.27	131.07	101.81	119.37	131.42	135.29	110.44	135.48	136.30	137.58	88.48	139.48	130.50	120.68	101.34	98.84
Phe+Tyr	60.00	120.90	113.33	137.32	120.73	113.63	118.32	137.24	103.24	123.21	126.44	134.87	119.61	147.34	157.79	151.91	112.20
Lys	55.00	131.13	144.64	122.83	136.91	145.02	150.36	142.39	151.30	139.01	132.22	102.85	161.01	124.84	118.22	105.00	81.13
Trp	10.00	163.54	178.95	1215.74	1172.36	148.43	101.00	1066.58	787.19	127.64	153.30	1138.86	37.31	141.45	183.78	1305.71	209.89

Notes: V, visceral mass; M, mantle; G, gill; A, adductor.

**Table 2 foods-11-02820-t002:** Analysis of protein composition in different parts of oyster (*Crassostrea hongkongensis*).

Tissue	WSP	SSP	ASP	ALSP
Visceral mass	Mb, cathepsin L, cathepsin V, albumen	MHC, MYL, A, TM	keratin I, MYL, peptide	tropoelastin, collagen, gelatin
Mantle	Mb, cathepsin L, cathepsin V, AAP, hemoglobin	MHC, A, PM	keratin I, collagen, gelatin, MYL	tropoelastin, collagen, gelatin
Gill	cathepsin V, AAP, hemoglobin	MHC, MYL, A, TM	MYL, peptide	gelatin
Adductor	cathepsin K, LAP, AAP, albumen, hemoglobin	MHC, MYL, PM, TM	keratin I, collagen, gelatin, MYL	collagen

Notes: Mb, myoglobin (16 kDa); Cathepsin L, 28 kDa; Cathepsin V, 35 kDa; Cathepsin K, 37 kDa; LAP, leucine aminopeptidase (96 kDa); AAP, alanine aminopeptidase (49.7 kDa); Albumen, 12 kDa; Hemoglobin, 64.5 kDa; MHC, myosin heavy chain (200 kDa); MYL, myosin light chain (16 kDa); A, actin (48 kDa); PM, paramyosin (100 kDa); TM, tropomyosin (30–40 kDa); Keratin I, 40–63 kDa; Collagen, >200 kDa; Gelatin, 50–100 kDa.

## Data Availability

The data presented in this study are available on request from the corresponding author.

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
