# Peer review of "The Spatial Distribution Patterns, Physicochemical Properties, and Structural Characterization of Proteins in Oysters (Crassostrea hongkongensis)"

_foods, 2022, doi:10.3390/foods11182820_

Round 1
Reviewer 1 Report
Review report
The spatial distribution patterns, physicochemical properties, and structural characterization of proteins in oyster (Crassostrea hongkongensis)
General
The authors have determined properties of protein fractions isolated from different parts of oyster. They have classified these proteins as WSP, SSP, ASP, and ALSP. Although they have carried out detailed analysis on the properties of different protein fractions, I have serious reservations regarding the methodology. In the first instance, the authors do not cite a reliable methodology for extractions of different fractions, they have mentioned. The work of Pan et al (ref.17), is on the isolation of a novel red colored pigment binding protein and is not relevant in this work. Secondly, the authors used trichloro acetic acid (#90) in the extraction, which is known to denature proteins resulting in loss of their natural structural and functional characteristics. The problem with methodology could have resulted in incorrect conclusion. In the Abstract *#23) the authors claim that the oyster had maximum water soluble proteins followed by SSP, ALSP, and ASP . This is against the literature. It is well known that seafood muscle contains 70 to 80% salt soluble myofibrillar proteins, followed by about 20% water soluble sarcoplasmic proteins. Therefore, the authors’ conclusion is erroneous.
Other comments:
There are mixing of sentences, lack of sequence and repetition, resulting in lack of flow of the contents in the Introduction section.
#38. Delete ‘with a year’
#63-64. Not clear
#66. ‘A discussion is required to describe what are (WSP), salt-soluble (SSP), acid-soluble (ASP) and alkaline-soluble (ALSP) are composed of?’. Are the myofibrillar proteins including myosin, actin, tropomyosin etc and connective tissue proteins collagen, gelatin etc. are there in these fractions? How much of these proteins are there in each of the component? A discussion on this aspect with available literature is required in the Introduction.
#153-154. This sentence & ref. are repletion
#69-71.How nutritional value can be determined by CD, FTIR, and fluorescence spectroscopy. Delete ‘nutritional value’ in the sentence.
#158.The reference is irrelevant to present discussion
#173 - #185. No information on the actual protein types in the selected parts of oyster. No discussion on the nature of proteins, except on SSP, which the authors identified as myofibrillar proteins (#183-184)
#173. WSP, SSP, ASP, and ALSP were the most common proteins. They may not be different proteins, but can be same proteins exhibiting different solubility characteristics. For example, water soluble proteins can also be salt soluble
#188. ‘The most significant aspect’ Modify as ‘One of the significant aspects’ .This is because digestibility is also equally important in determining protein quality
#191. Can you please spell out the TAA?
#207. Oyster proteins contain up to 40%-50% of flavor amino acids, thus suggesting that oysters have a more pleasant flavor. 40-50^of what? Of the total amino acids in the proteins?
#211-213.Reference for FAO? The sentence is not clear also..
#414. A better sentence is ‘the protein content was maximum in visceral mass followed by mantle, adductor, and gill.
#416.What are flavor amino acids present in oyster? Not mentioned in the manuscript
The the molecular weight distribution of SSP was wider (16-270 kDa), The broad range can be due to the presence of more than one type of proteins. The fraction may contain not only the large protein, myosin, but also smaller ones
References:
#3 The paper is on mushroom. It does not match with the present work. A paper related to seafood is suggested
#52. Is not relevant
#211-213. Reference for FAO not provided
Please try to keep references related to oyster and other fishery products.
Author Response
General
The authors have determined properties of protein fractions isolated from different parts of oyster. They have classified these proteins as WSP, SSP, ASP, and ALSP. Although they have carried out detailed analysis on the properties of different protein fractions, I have serious reservations regarding the methodology.
(1) In the first instance, the authors do not cite a reliable methodology for extractions of different fractions, they have mentioned. The work of Pan et al (ref.17), is on the isolation of a novel red colored pigment binding protein and is not relevant in this work.
Response: Pan et al. extracted WSP, SSP, ASP and ALSP according to the solubility of protein, which are consistent with the purpose of extraction in this study.
2.2.2. Non-water-soluble protein
In addition to the traditional water-soluble proteins, non-water soluble proteins were also scanned for the existence of red colour-related proteins. They were divided into salt-, acid-, and alkaline-soluble parts based on protein solubility and extracted according to Fernlund's method (Fernlund & Josefsson, 1968). 30% interfering precipitate was re-dissolved into 50 mM phosphate buffer containing 1 M NaCl and the salt-soluble extract was collected by centrifugation at 15,000×g for 20 min at 4°C. The precipitate was soaked in 0.1 M HCl or NaOH to extract acidic- and alkaline-soluble proteins, respectively. All precipitate obtained from the previous step was washed with distilled water (pH 7.0) three times before being subjected to subsequent extraction experiments.
(2) Secondly, the authors used trichloro acetic acid (#90) in the extraction, which is known to denature proteins resulting in loss of their natural structural and functional characteristics.
Response: The experimental method was not clearly written. In reality, WSP was extracted first, and then part of the WSP solution was taken and trichloroacetic acid was added, in order to remove the non-protein nitrogen and avoid the effect of the presence of non-protein nitrogen on the water-soluble protein content. The proteins used in the subsequent experiments were not treated with TCA, and the extraction method has been corrected in the paper.
(3) The problem with methodology could have resulted in incorrect conclusion. In the Abstract *#23) the authors claim that the oyster had maximum water-soluble proteins followed by SSP, ALSP, and ASP. This is against the literature. It is well known that seafood muscle contains 70 to 80% salt soluble myofibrillar proteins, followed by about 20% water soluble sarcoplasmic proteins. Therefore, the authors’ conclusion is erroneous.
Response: The results of this study were similar to those of Zhang et al. 2020 in " Physicochemical state and in vitro digestibility of heat treated water-soluble protein from Pacific oyster (Crassostrea gigas)" in Food Bioscience regarding WSP and SSP. Zhang et al. showed that the contents of WSP and SSP accounted for 38% and 31% of crude protein in oysters, respectively. Karnjanapratum et al. published in Food Chemistry in 2013 "Chemical compositions and nutritional value of Asian hard clam (Meretrix lusoria) from the coast of Andaman Sea", the results showed that Asian hard clam (Meretrix lusoria) was high in ALSP, followed by WSP and SSP.
The myofibrillar proteins in general seafood muscles cannot reach 70-80%. For example, fish have 60-75% myofibrillar proteins, 20-35% sarcoplasmic proteins and 2%-20% myostromins in their muscles.
Zou, H.; Zhao, N.; Li, S.; Sun, S.; Dong, X.; Yu, C. Physicochemical and emulsifying properties of mussel water-soluble proteins as affected by lecithin concentration. International Journal of Biological Macromolecules 2020, 163, 180-189. DOI: 10.1016/j.ijbiomac.2020.06.225.
Karnjanapratum, S.; Benjakul, S.; Kishimura, H.; Tsai, Y. H. Chemical compositions and nutritional value of Asian hard clam (Meretrix lusoria) from the coast of Andaman Sea. Food Chem 2013, 141
(4), 4138-4145. DOI: 10.1016/j.foodchem.2013.07.001.
Other comments: (1) There are mixing of sentences, lack of sequence and repetition, resulting in lack of flow of the contents in the Introduction section.Response: The introductory section has been revised in the revised manuscript. (2) #38. Delete ‘with a year’
Response: Done.
(3) #63-64. Not clear
Response: Oysters have been mainly sold as raw, dried products, condiments and canned foods. In recent years, oyster functional products, nutritional supplements and other high value-added protein products have received wide attention. Therefore, the study of oyster proteins can provide the necessary theoretical basis for the processing and preservation and high-value utilization of oysters.
Considering that this paragraph does not fit well here, it was deleted in the revised manuscript.
(4) #66. ‘A discussion is required to describe what are (WSP), salt-soluble (SSP), acid-soluble (ASP) and alkaline-soluble (ALSP) are composed of?’. Are the myofibrillar proteins including myosin, actin, tropomyosin etc and connective tissue proteins collagen, gelatin etc. are there in these fractions? How much of these proteins are there in each of the component? A discussion on this aspect with available literature is required in the Introduction.
Response: The WSP, SSP, ASP, and ALSP were added to the introduction of the revised manuscript.
(5) #153-154. This sentence & ref. are repletion
Response: References have been inserted in the revised manuscript.
(6) #69-71.How nutritional value can be determined by CD, FTIR, and fluorescence spectroscopy. Delete ‘nutritional value’ in the sentence.
Response: Done.
(7) #158.The reference is irrelevant to present discussion
Response: References have been revised in the revised manuscript.
(8) #173 - #185. No information on the actual protein types in the selected parts of oyster. No discussion on the nature of proteins, except on SSP, which the authors identified as myofibrillar proteins (#183-184)
Response: This part has been added to the revised manuscript.
(9) #173. WSP, SSP, ASP, and ALSP were the most common proteins. They may not be different proteins, but can be same proteins exhibiting different solubility characteristics. For example, water soluble proteins can also be salt soluble.
Response: There is some truth to this statement. For example, a small percentage of WSP or SSP may have both water-soluble and salt-soluble properties, but the primary solubility properties of WSP and SSP are water-soluble and salt-soluble, respectively.
(10) #188. ‘The most significant aspect’ Modify as ‘One of the significant aspects’. This is because digestibility is also equally important in determining protein quality.
Response: Done.
(11) #191. Can you please spell out the TAA?
Response: TAA is an abbreviation for total amino acids and has been added to the revised manuscript.
(12) #207. Oyster proteins contain up to 40%-50% of flavor amino acids, thus suggesting that oysters have a more pleasant flavor. 40-50^of what? Of the total amino acids in the proteins?
Response: Flavor amino acids are Asp, Glu, Gly, Ala, Tyr, and Phe, respectively. 40-50% is the percentage of flavor amino acids in the total amino acids.
(13) #211-213.Reference for FAO? The sentence is not clear also.
Response: Added to the FAO/WHO (1973) recommended model in the revised manuscript.
“According to the WHO/FAO recommended model, the ratio of essential amino acids to total amino acids (EAA/TAA) is about 40% and the ratio of essential amino acids to non-essential amino acids (EAA/NEAA) is more than 60% for proteins of good quality.”
(14) #414. A better sentence is ‘the protein content was maximum in visceral mass followed by mantle, adductor, and gill.
Response: Thanks very much for the suggestion. This sentence has been revised in the revised manuscript.
(15) #416.What are flavor amino acids present in oyster? Not mentioned in the manuscript
Response: The flavor amino acids in oysters include Asp, Glu, Gly, Ala, Tyr, Phe, which have been added in the revised manuscript.
(16) The molecular weight distribution of SSP was wider (16-270 kDa), The broad range can be due to the presence of more than one type of proteins. The fraction may contain not only the large protein, myosin, but also smaller ones.
Response: Thanks very much for the suggestion. This content has been added to the revised manuscript.
References:
(17) #3 The paper is on mushroom. It does not match with the present work. A paper related to seafood is suggested
Response: This reference has been removed in the revised manuscript.
(18) #52. Is not relevant
Response: This reference has been removed in the revised manuscript.
(19) #211-213. Reference for FAO not provided
Response: This reference has been added in the revised manuscript.
Please try to keep references related to oyster and other fishery products.
Response: Thanks very much for the suggestion. Reference improvements have been made in the revised manuscript.

Reviewer 2 Report
Authors investigated the spatial distribution (visceral mass, mantle, gill, and adductor muscle) patterns and structural characteristics of proteins, including water-soluble (WSP), salt-soluble (SSP), acid-soluble (ASP), and alkaline (ALSP) of oyster with the aim to provide an essential reference for the processing and preservation of oysters and explore new ways to develop and apply high value-added oysters.
This manuscript is written well and tried to use some innovative methods besides common chemical analysis of proteins. However, I do believe that the review in its current form has flaws that need to be addressed thus major revisions should be done.
First, authors used oyster that is purchased from a local fish store in one city. If authors aim to provide a reference value for oyster processing or production of high-value ingredients, sampling also should be done across a wide geographical region. Different places of coastal waters have different physicochemical properties, temperature, bottom type, or associated planktonic community. These physicochemical parakeets of the water where oyster growth will influence on physicochemical parameters of proteins in oyster. On the other hand, parameters related to size or weight of oysters influence oyster proteins and flavor, manuscript also lacks the investigating the role of oyster size. Unfortunately, authors did not consider the role of environment in their study (i.e., sampling different oysters from different area). So, the obtained results are difficult to be generalized for all oysters. Please explain?
Introduction should be revised hugely according to above comment, in the current form it discusses some general findings on oyster, rather than should be focus on actual objectives of the study. Authors should discuss on different types of proteins they are speaking in their manuscript including salt- or water-soluble, their role in post-harvest quality and processing.
Authors investigating the proteins of edible and non-human edible parts of oyster, such as viscera. It is obvious that viscera are more susceptible to oxidation if oysters handled or transported improperly. Please explain? In addition, what “high value-added oyster’s products” authors mean or suggested?
Author Response
(1) First, authors used oyster that is purchased from a local fish store in one city. If authors aim to provide a reference value for oyster processing or production of high-value ingredients, sampling also should be done across a wide geographical region. Different places of coastal waters have different physicochemical properties, temperature, bottom type, or associated planktonic community. These physicochemical parakeets of the water where oyster growth will influence on physicochemical parameters of proteins in oyster. On the other hand, parameters related to size or weight of oysters influence oyster proteins and flavor, manuscript also lacks the investigating the role of oyster size. Unfortunately, authors did not consider the role of environment in their study (i.e., sampling different oysters from different area). So, the obtained results are difficult to be generalized for all oysters. Please explain?
Response: The oysters Crassostrea hongkongensis were purchased from a large wholesale fish market in Zhanjiang in December. The oysters were of medium size, with a length of 12.8±0.6 cm and a width of 6.0±0.4 cm, 50 kg of oysters in the shell yielded about 6-7 kg of oyster meat. A total of 400 kg of oysters in the shell were purchased in four times. The oysters used in the experiment were basically representative of the oyster quality in Zhanjiang. When designing the experimental protocol, we had considered a comparative study of oyster proteins from different farming bases. However, since we did not have a cooperative farming base at that time, the experimental raw materials had to be purchased from the market, thus we could not ensure the accuracy of the oyster origin, so we had to abandon this research idea. Meanwhile, Qin et al. published a paper in Food Chemistry in 2021 entitled "Seasonal variations in biochemical composition and nutritional quality of Crassostrea hongkongensis, in relation to the gametogenic cycle”. Studies have shown that the protein content of the various tissues in Crassostrea hongkongensis exceeded 52% of the dry weight at different times of the year, with the only significant change in protein levels was an increase between April and June, and the optimal harvest time of Crassostrea hongkongensis was from August to February when oyster quality and taste were at their best. Many researchers have found that the timing and period of spawning and the duration of gametogenesis in oyster varied between locations, and have attributed this to latitude, temperature, salinity and food availability. However, the relatively narrow range of distribution of Crassostrea hongkongensis, which is confined to estuarine areas of southern China. Therefore, the overall trend of the growth cycle of Crassostrea hongkongensis is consistent. So, the present study has some reference value for the processing and utilization of Crassostrea hongkongensis.
(2) Introduction should be revised hugely according to above comment, in the current form it discusses some general findings on oyster, rather than should be focus on actual objectives of the study. Authors should discuss on different types of proteins they are speaking in their manuscript including salt- or water-soluble, their role in post-harvest quality and processing.
Response: The introductory section has been revised in the revised manuscript as required.
(3) Authors investigating the proteins of edible and non-human edible parts of oyster, such as viscera. It is obvious that viscera are more susceptible to oxidation if oysters handled or transported improperly. Please explain? In addition, what “high value-added oyster’s products” authors mean or suggested?
Response: Aquatic animal muscles contain a variety of proteases, and the main enzymes that play an important role in muscle protein degradation and free amino acid accumulation are aminopeptidase, trypsin-like serine proteinase, and cathepsins. Compared with the mantle, gill and adductor, the viscera have high protease content and has the highest enzyme activity. Therefore, viscera are more susceptible to autolysis.
We have studied the proteins in different parts of oyster (visceral mass, mantle, gill and adductor) in order to fully understand the types, distribution and physicochemical properties of oyster proteins. The aim of this study is to exploit the composition and properties of oyster proteins to develop high value-added products. For example, the high metal content of gill can be used to develop Zinc-binding protein supplements; the high protease content and enzyme activity of visceral mass can develop protease products; the high water-soluble protein content of oyster can develop functional protein peptide products; High flavor amino acid content can be developed into condiments, etc.

Round 2
Reviewer 1 Report
Manuscript on oyster (The spatial distribution patterns……Li et al.,)
The paper makes detailed and exhaustive studies on the physic-chemical and structural characteristics of proteins isolated from different parts of oyster. These protein isolates include water soluble proteins (WSP), salt soluble proteins (SSP), acid soluble proteins (ASP) and alkali soluble proteins (ALSP).
My concerns on the manuscript are:
(i) How water soluble proteins can be the most abundant class of proteins in oyster, as the authors claim? This is against conventional literature on the nature of proteins of fish/shellfish muscle, and, no work is cited to substantiate this claim.
(ii) What is the significance of the work? How the information on the characteristics of proteins soluble in different extractants can help processing and preservation of oysters and explore new ways to develop and apply high value-added oysters, as the authors conclude.
(iii) I suggest the authors clarify these points. A detailed 9preferably with a Table) on the types of proteins such as myosin, actomyosin, elastin, collagen, enzymes and others that can be present in different extracts may be desirable
2#28-29. Oyster proteins (e.g., WSP, SSP, ASP and ALSP) have different physicochemical, functional and structural properties, and are suitable for different processing methods {13-17). The cited references discuss effects of high pressure treatment on oyster proteins; they do not discuss WSP, SSP, ASP and ALSP. Only the paper by Yu et al (Ref. 14) discusses effect of high pressure on acid treated oyster protein isolate
Editorial corrections are required in many places of the manuscript. Some are pointed out below:
Page 2, Line # 11. The oyster's color, taste, and nutritional quality were significantly improved? Not clear
2#12. Delete ‘that the only’
2#13. Delete ‘was an increase’
2#20. Digestion. Proteolytic digestion?
#32-33. Unlike most shellfish whose muscle protein composition is dominated by SSP, oysters are dominated by WSP, followed by SSP. No reference also given to support the statement
3/20. SSP was prepared according to Zhang et al. with slight modifications [8]. Elaborate on the methodology
5/50 ‘SSP in different parts of the oysters’. Do the authors mean that SSP is present in different parts of oysters?
One useful reference is:
Mazorra-Manzano, M. A., Ramírez-Suárez, J. C.,Moreno-Hernández, J. M.,and. Pacheco-Aguilar, R. (2018) Seafood proteins, in Proteins in Food Processing, 2nd ed. R. Y. Yada (ed.) Woodhead publ.,Ch. 17, pp.445-475
Author Response
The paper makes detailed and exhaustive studies on the physic-chemical and structural characteristics of proteins isolated from different parts of oyster. These protein isolates include water soluble proteins (WSP), salt soluble proteins (SSP), acid soluble proteins (ASP) and alkali soluble proteins (ALSP).
My concerns on the manuscript are:
(i) How water-soluble proteins can be the most abundant class of proteins in oyster, as the authors claim? This is against conventional literature on the nature of proteins of fish/shellfish muscle, and, no work is cited to substantiate this claim.
Response: After several repeated experiments, we determined that the water-soluble protein content in oysters was higher than the salt-soluble protein. The results of this study were similar to those of the study "The separation and composition of oyster protein" published by Jingjing Zhang in Food and Fermentation Industries. The water-soluble protein content of oysters is higher than the salt-soluble protein, which may be due to the following reasons: The textural characteristics of oyster meat: tender, juicy, soft, and significantly lower in hardness value and elasticity than the muscle tissue of most aquatic products; oyster meat precipitates a large amount of milky liquid after steaming, which is mainly water-soluble protein. Therefore, the water-soluble protein in oyster meat is indeed the highest.
We have inserted references in the revised manuscript to prove this claim.
Zhang, J. J., Zheng, H. N., Zhang, C. H., Hao, J. M., Zhang, J., & Zhang, J. Y. (2013). The separation and composition of oyster protein. Food and Fermentation Industries, 39(9), 195-199.
Zou, H.; Zhao, N.; Li, S.; Sun, S.; Dong, X.; Yu, C. Physicochemical and emulsifying properties of mussel water-soluble proteins as affected by lecithin concentration. International Journal of Biological Macromolecules 2020, 163, 180-189. DOI: 10.1016/j.ijbiomac.2020.06.225.
(ii) What is the significance of the work? How the information on the characteristics of proteins soluble in different extractants can help processing and preservation of oysters and explore new ways to develop and apply high value-added oysters, as the authors conclude.
Response: This study was conducted to explore the correlation between protein characteristics and their edible quality through the study of several different protein characteristics in various parts of oyster muscles, to enrich the basic research data of aquatic products in China, and to provide relevant theoretical support and practical basis for the deep processing of oysters in the future. Due to the variability of different parts of oyster muscles, we choose suitable processing methods to reduce the nutritional loss in the future processing process. At the same time, new products can be developed in the industry for the different hardness and toughness of the muscles in different locations and for the consumption habits of different people (old people, children and young people). In addition, in view of the protein variability of different parts of oysters, this study provides new ideas for the use of preservatives (type, location and method) and the development of high value-added products (protease, condiments, nutritional health products, snack foods, etc.) in the processing of oyster frozen products in the future.
(iii) I suggest the authors clarify these points. A detailed 9preferably with a Table) on the types of proteins such as myosin, actomyosin, elastin, collagen, enzymes and others that can be present in different extracts may be desirable.
Response: We add the corresponding figure and table to the revised manuscript.
Table 2. Analysis of protein composition of different parts of oyster (Crassostrea hongkongensis).
|
Tissue |
WSP |
SSP |
ASP |
ALSP |
|
Visceral mass |
Mb, cathepsin L, cathepsin V, albumen |
MHC, MYL, A, TM |
keratin â… , MYL peptide |
tropoelastin, collagen, gelatin |
|
Mantle |
Mb, cathepsin L, cathepsin V, AAP, hemoglobin |
MHC, A, PM |
keratin â… , collagen, gelatin, MYL |
tropoelastin, collagen, gelatin |
|
Gill |
cathepsin V, AAP, hemoglobin |
MHC, MYL, A, TM |
MYL, peptide |
gelatin |
|
Adductor |
cathepsin K, LAP, AAP, albumen, hemoglobin |
MHC, MYL, PM, TM |
keratin â… , collagen, gelatin, MYL |
collagen |
Notes: Mb, myoglobin (16 kDa); Cathepsin L, 28 kDa; Cathepsin V, 35 kDa; Cathepsin K, 37 kDa; LAP, leucine aminopeptidase (96 kDa); AAP, alanine aminopeptidase (49.7 kDa); Parvalbumin, 12 kDa; Hemoglobin, 64.5 kDa; MHC, myosin heavy chain (200 kDa); MYL, myosin light chain (16 kDa); A, actin (48kDa); PM, paramyosin (100 kDa); TM, tropomyosin (30-40 kDa); Keratin â… , 40-63 kDa; Collagen, >200 kDa; Gelatin, 50-100kDa.
(1) 2#28-29. Oyster proteins (e.g., WSP, SSP, ASP and ALSP) have different physicochemical, functional and structural properties, and are suitable for different processing methods {13-17). The cited references discuss effects of high pressure treatment on oyster proteins; they do not discuss WSP, SSP, ASP and ALSP. Only the paper by Yu et al (Ref. 14) discusses effect of high pressure on acid treated oyster protein isolate.
Response: We have reinserted the references in the revised manuscript.
(2) Editorial corrections are required in many places of the manuscript. Some are pointed out below:
Page 2, Line # 11. The oyster's color, taste, and nutritional quality were significantly improved? Not clear
Response: This sentence was expressed incorrectly and has been corrected in the revised manuscript.
“Diploidand and triploid Crassostrea hongkongensis were high protein foods that maintained relatively stable levels and a relatively well-balanced composition of essential amino acids between the reproductive and non-reproductive phases. This suggested that dip-loid and triploid Crassostrea hongkongensis were high quality sources of protein.”
(3) 2#12. Delete ‘that the only’
Response: The words has been removed from the revised manuscript.
(4) 2#13. Delete ‘was an increase’
Response: The words has been removed from the revised manuscript.
(5) 2#20. Digestion. Proteolytic digestion?
Response: WSP of different parts of the oyster simulate gastrointestinal digestion, the ease of digestion was visceral mass, gill, mantle and adductor in descending order.
We have made additions in the revised manuscript.
(6) #32-33. Unlike most shellfish whose muscle protein composition is dominated by SSP, oysters are dominated by WSP, followed by SSP. No reference also given to support the statement.
Response: We have inserted references in the revised manuscript.
(7) 3/20. SSP was prepared according to Zhang et al. with slight modifications [8]. Elaborate on the methodology.
Response: We have added the detailed steps of SSP extraction into the revised manuscript.
“The four times volume of 0.1 M PBS (pH 7.2, 0.5 M NaCl) was added to the remaining precipitate after extraction of the water-soluble proteins, and then stirred at 4 â—¦C for 18 h. The mixture was centrifuged at 12000 rpm for 20 min (4 â—¦C) and the supernatant was collected. The extraction steps were repeated three times. The three supernatants were combined and dialyzed with the 3.5 kDa cut-off dialysis tubing. After that all of the supernatants were lyophilized to obtain the protein powder. ”
(8) 5/50 ‘SSP in different parts of the oysters’. Do the authors mean that SSP is present in different parts of oysters?
Response: No, that's not the point we were trying to make. What we want to say is that " SSP is the most abundant in the adductor, followed by the mantle, visceral mass and gill."
(9) One useful reference is:
Mazorra-Manzano, M. A., Ramírez-Suárez, J. C.,Moreno-Hernández, J. M.,and. Pacheco-Aguilar, R. (2018) Seafood proteins, in Proteins in Food Processing, 2nd ed. R. Y. Yada (ed.) Woodhead publ.,Ch. 17, pp.445-475
Response: Thank you very much for your help and we have inserted this reference in the revised manuscript.

Reviewer 2 Report
Authors responded to comments adequately.
Author Response

(The authors gave the same response as above.)
